# Comparison of Cefazolin/Metronidazole to Ampicillin/Sulbactam as Preoperative Antibiotics in Colorectal Surgery: A Retrospective, Single-Center Cohort Study

**DOI:** 10.3390/antibiotics12091381

**Published:** 2023-08-29

**Authors:** Jae Hyoung Im, Dong Yeop Lee, Ji Hyeon Baek, Se Ju Lee, Sungtaek Jung, Eunjung Kim, Dong Yoon Kang, Jin-Soo Lee

**Affiliations:** 1Division of Infectious Diseases, Department of Internal Medicine, Inha University College of Medicine, Incheon 22212, Republic of Korea; idjh@inha.ac.kr (J.H.I.); jhbaek@inha.ac.kr (J.H.B.); playit@nate.com (S.J.L.); 2Department of Preventive Medicine, Ulsan University Hospital, Ulsan 44033, Republic of Korea; eastside323@naver.com; 3Department of General Surgery, Shihwa Hospital, Siheung 15034, Republic of Korea; habana95@hanmail.net; 4Infection Control Unit, Inha University Hospital, Incheon 22212, Republic of Korea; eunjungii@hanmail.net

**Keywords:** ampicillin, antibiotics prophylaxis, cefazolin, colorectal surgery

## Abstract

Aim: The use of prophylactic antibiotics prior to colorectal surgery reduces surgical site infections. Cefazolin and metronidazole are used as a standard regimen. Ampicillin/sulbactam may be an alternative, but current data are limited. We compared the efficacy of ampicillin/sulbactam with cefazolin and metronidazole as prophylactic antibiotics. Methods: Patients who underwent colorectal surgery at Inha University Hospital between 2010 and 2020 were treated prophylactically with cefazolin and metronidazole or ampicillin/sulbactam, and observed for 30 days following surgery. The primary outcome was surgical site infections. The secondary outcomes were deep/organ infections and the need for drainage. Results: SSIs occurred in 2.6% (17/646) of the ampicillin/sulbactam group, whose rate was not inferior to the occurrence in the group receiving cefazolin and metronidazole (3.8%, 21/556). There was no significant difference between the two groups in the secondary outcomes. Conclusions: Compared to the cefazolin and metronidazole combination, ampicillin/sulbactam is not inferior as a preoperative prophylactic antibiotic regimen for colorectal surgery.

## 1. Introduction

Surgical site infections (SSIs) are one of the main causes of nosocomial infections, which lower the quality of life, lengthen hospital stay, and increase the economic burden [1]. SSIs also increase the chances of hospital readmission by five times and death by two times [2]. Many studies have sought to reduce SSIs. Antibiotics administered prophylactically before surgery significantly lower this risk [3]. In particular, because colorectal surgery resects the intestine, the risk of infection is higher than in other clean surgeries [4]. SSIs occur in approximately 4–10% of patients undergoing colon surgery and in 3–27% of patients after rectal surgery [5]. In colorectal surgery, pathogens mainly originate in the intestine [6]. Therefore, antibiotics that act on Enterobacteriaceae and anaerobes are selected as preoperative antibiotics [7].

Several preoperative antibiotic options may be available. Antibiotics such as carbapenem have been used in institutions reporting severely resistant strains [7]. The most commonly used prophylactic antibiotic regimen is a combination of first-generation cephalosporin and metronidazole [8,9]. Ampicillin/sulbactam is easily administered and is active against *Enterococcus*. Therefore, this combination may be an alternative to the combination of first-generation cephalosporin and metronidazole. However, studies on the effectiveness of ampicillin/sulbactam as a prophylactic antibiotic were relatively outdated and involved a small number of patients [10]. It is possible that antibiotic susceptibility and gut flora have changed. Thus, more recent data are needed. Therefore, we planned a study to compare the effectiveness of cefazolin/metronidazole and ampicillin/sulbactam as prophylactic antibiotics before surgery.

## 2. Results 

### 2.1. General Characteristics

A total of 556 patients were assigned to the CFZ/MTZ group and 646 to the AMP/SUL group. The mean age of the patients was 64.3 years, and 56% were male. There was no statistically significant difference in most variables between the two groups. However, the rates of rectal surgery and obesity were higher in the CFZ/MTZ group (Table 1).

### 2.2. Risk Analysis of SSI Using Logistic Regression Analysis

Smoking was the most significant risk factor, with an odds ratio (OR) of 3.489 [95% confidence interval (CI): 1.784–6.8222]. Long operation time and male sex appeared to be risk factors in the univariate analysis, but there were no statistically significant differences in the multivariate analysis. Compared to the AMP/SUL group, SSI was more frequent in the CFZ/MTZ group, but the difference was not statistically significant in either the univariate or multivariate analysis (Table 2).

### 2.3. Comparison of Outcomes of the Two Groups (t-Test, Non-Inferiority)

SSIs occurred in 2.6% (17/646) of the AMP/SUL group, whose rate was not inferior to the rate of 3.8% (21/556) in the CFZ/MTZ group (*p* = 0.264, CI= −0.009–0.032). Deep/organ space infections occurred in 1.24% (8/646) of the AMP/SUL group, whose rate was non-inferior to the rate of 2.7% (15/556) in the CFZ/MTZ (*p* = 0.073, CI= −0.001–0.031). The need for drainage was also non-inferior, with values of 0.6% (4/646) in the AMP/SUL group and 1.6% (9/556) in the CFZ/MTZ group (*p* = 0.167, CI= −0.003–0.020) (Table 3 and Figure 1). In the CFZ/MTZ group, SSIs involved *Enterococcus* spp. (*n* = 3), *Escherichia coli* (*n* = 5), *Klebsiella pneumoniae* (*n* = 1), *Pseudomonas* (*n* = 1), *Morganella morganii* (*n* = 1), and methicillin-resistant *Staphylococcus aureus* (MRSA, *n* = 2). SSIs in the AMP/SUL group involved *Enterococcus* spp. (*n* = 2), *Escherichia coli* (*n* = 2), and Enterobacter (*n* = 1). Extended-spectrum β-lactamase (ESBL)-producing pathogens were isolated from one case in the AMP/SUL group and three cases in the CFZ/MTZ group.

## 3. Discussion

There have been few studies on the effects of ampicillin/sulbactam as a preoperative antibiotic in colorectal surgery. In a previous study conducted in the 1990s, a comparison of patients treated with the combination of ampicillin/sulbactam (*n* = 63) and patients treated with gentamicin and metronidazole (*n* = 65) did not show a significant difference in the incidence of SSIs (9.5% vs. 10.7%) [10]. Another paper in the 1990s compared amoxycillin/clavulanate to cefotaxime + metronidazole as prophylactic antibiotic regimens in colorectal surgery. No difference in the effectiveness of preventing SSIs was evident between the two regimens, with SSIs occurring in 8 out of 76 patients treated with amoxycillin/clavulanate and 9 out of 88 patients treated with cefotaxime + metronidazole [11]. Based on these studies, ampicillin/sulbactam has been suggested as an alternative to preoperative antibiotics in colorectal surgery. However, this has not been established definitively.

A 2013 study analyzed the effects of prophylactic antibiotics In patients who underwent colorectal surgery at 112 hospitals. The subgroups of the study included ampicillin/sulbactam and cefazolin + metronidazole [12]. In that study, SSIs occurred in 11.3% of the 363 patients treated with ampicillin/sulbactam. This rate was inferior to the rate of 6.2% in the 899 patients treated with cefazolin/metronidazole. In the above-mentioned study, the possibility of selecting antibiotics according to the preoperative wound class could not be excluded. In addition, there was a strong correlation between SSIs and the skills of the surgeon, which could not be controlled. Furthermore, the number of patients in the ampicillin/sulbactam group was relatively small. In contrast, our study involved only specialists, excluding those with <10 years of experience. Moreover, in our study, antibiotic selection was distributed by period. Thus, the possibility of bias in the selection of preoperative antibiotics according to the surgeon’s propensity or preoperative wound class was low.

As a rationale for the inferior prophylactic effect of ampicillin/sulbactam compared to cefazolin + metronidazole in the aforementioned study [13], the authors suggested increasing antibiotic resistance of *Bacteroides fragilis*. Increasing resistance of *B. fragilis to* ampicillin/sulbactam has been evident since 2010. However, the most recently reported nosocomial intra-abdominal infections are *Escherichia coli*, *Pseudomonas* spp., and *Enterococcus* spp. [13,14]. Similarly, in our study, anaerobes were not isolated in the ampicillin/sulbactam group. Therefore, there is no evidence that ampicillin/sulbactam is more vulnerable to anaerobe-induced SSIs than cefazolin and metronidazole.

An increase in the viable count of *Enterococcus* should be considered in the selection of prophylactic antibiotics for colorectal surgery. *Enterococcus* infection is a major complication of colorectal surgery [15] and can cause anastomotic leakage [16]. Ampicillin is the preferred therapeutic agent for susceptibility to *Enterococcus* species [17]. In contrast, cefazolin is intrinsically resistant to cephlosporins in *Enterococcus* species [17]. Changes in antibiotic susceptibility should also be considered. The increasing number of ESBL-producing pathogens is emerging as a major problem in nosocomial infections. One study reported that 38% of SSIs after colorectal surgery were caused by ESBL-producing pathogens [18]. Sulbactam is a β-lactamase inhibitor that is beneficial for ESBL-producing strains [19]. Considering this pattern of resistance, changes may be necessary in the selection of prophylactic antibiotics. In our study, deep/organ infections tended to be lower in the AMP/SUL group than in the CFZ/MTZ group, although the difference was not statistically significant. In cultured bacteria, Enterobacteriaceae, including *Escherichia coli*, were less abundant in the AMP/SUL group. ESBL production was also not statistically significantly different, although it was lower in the AMP/SUL group than in the CFZ/MTZ group. Considering the increasing resistance of Enterobacteriaceae to antibiotics, AMP/SUM appears suitable as a preoperative antibiotic regimen for colorectal surgery.

This study has several limitations. First, it was a retrospective study with a small number of patients. Second, it was not possible to control all variables, such as the handwashing performance rate and environmental disinfection. Third, differences in side effects, such as gastrointestinal trouble or *Clostridium difficile*-associated disease, were not investigated. Fourth, there might be differences in the susceptibility of pathogens to antibiotics by region. Fifth, skin and bowel preparations were performed for all patients who underwent colorectal surgery during the study period. Although such preparations are considered not recommended or controversial, they could not influence the results of the study. Sixth, data on laparotomy history, oxygen administration, and blood transfusion could not be obtained. These variables should be better controlled in future studies to minimize variability.

In conclusion, ampicillin/sulbactam is not inferior to cefazoline/metronidazole as a prophylactic antibiotic regimen for colorectal surgery. It is necessary to consider the susceptibility to antibiotics by region and institution. Large-scale studies controlling for various variables are needed.

## 4. Materials and Methods

### 4.1. Study Population

Patients who underwent colorectal surgery at Inha University Hospital between July 2010 and December 2020 were enrolled. Patients treated with emergency surgery or combined surgery with other organs were excluded from this study. Specialists with <10 years of experience in colorectal surgery were excluded. Only patients who underwent surgery from three specialists were analyzed. These patients were divided into two groups according to preoperative antibiotics: (i) cefazolin (intravenous, 2 g) and metronidazole (intravenous or oral, 500 mg) combination (CFZ/MTZ group), and (ii) ampicillin/sulbactam (intravenous, 3 g; AMP/SUL group).

### 4.2. Characteristics of Patients

The following indicators were used to compare the characteristics of patients undergoing surgery: age, sex, operation time, wound class, American Society of Anesthesiologists (ASA) score, endoscopic operation, diabetes, obesity, and smoking. The preoperative wound class was divided into four levels: clean, clean-contaminated, contaminated, and dirty. ASA scores were divided into six stages: healthy, with mild systemic diseases, severe systemic diseases, diseases that are a constant threat to life, moribund patients who are not expected to survive without the operation, and declared brain death. Obesity was defined as a body mass index ≥ 30 kg/m^2^.

### 4.3. Antibiotic Prophylaxis Protocol

All patients who underwent colorectal surgery during the study period underwent bowel and skin preparations. Antibiotics were administered at least 1 h prior to skin incision, and re-dosing was conducted 4 h after the initial dose was administered. No antibiotics were administered after the 24 h post-surgery. The type of antibiotic used was based on the period, whereby AMP/SUL was used from 2011 to 2013 and from 2016 to 2020, while CFZ/MTZ was used during the other periods. The Infection Control Unit and infectious medicine specialist monitored the preoperative antibiotic protocols and antibiotic stewardship, while the Department of General Surgery managed other surgical preparations.

### 4.4. Outcomes

SSIs in the 30 days following surgery were the primary outcome. SSIs were defined according to the criteria of the Centers for Disease Control and Prevention [20]. The secondary outcomes were the grade of SSI and treatment method. The SSI grade was (i) none, (ii) superficial incisional, (iii) deep incisional, and (iv) organ/space. The treatment methods were divided into (i) only antibiotics and (ii) antibiotics with percutaneous or surgical drainage.

### 4.5. Ethics Statement

This study was approved by the Institutional Review Board of Inha University Hospital, Incheon, Korea. All patient records were anonymized.

### 4.6. Statistical Analyses

Chi-square test or Fisher’s exact test was used to compare the general characteristics of the CFZ/MTZ and AMP/SUL groups. A non-inferiority test was performed on the incidence of SSIs between the CFZ/MTZ and AMP/SUL groups. The margin was set at 5%. Logistic regression analysis was used to analyze the risk of SSIs. For logistic regression analysis, wound classes were divided into clean or clean/contamination and contaminated/dirty. The ASA score was divided into 1–2 points and 3–6 points. The operation time was divided based on the time required for 75% of the total colorectal surgeries. A two-sided test was used, and a *p*-value ≤ 0.05 was considered statistically significant. The analyses were performed using SPSS version 21.0.

## Figures and Tables

**Figure 1 antibiotics-12-01381-f001:**
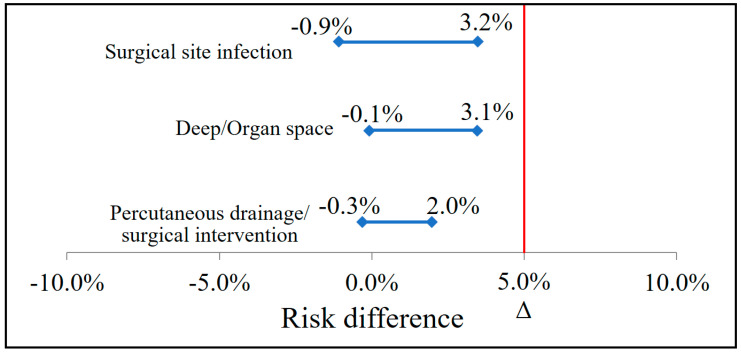
Non-inferiority test of prophylactic effect of AMP/SUL compared to CFZ/MTZ. Δ = non-inferiority margin AMP/SUL therapy was not inferior to CFZ/MTZ therapy in prophylactic effect at a margin of 5%.

**Table 1 antibiotics-12-01381-t001:** Comparison of general characteristics of study populations.

Variables	CFZ/MTZ (*n* = 556)	AMP/SUL (*n* = 646)	*p*-Value
Age in years, mean ± standard deviation	64.2 ± 12.0	64.4 ± 12.8	0.831
Sex			
Male	318 (47.3)	355 (52.7)	0.435
Female	238 (45.0)	291 (55.0)	
Surgical site			
Colon	267 (43.1)	353 (56.9)	0.019
Rectum	289 (49.8)	291 (50.2)	
Operation time			
<75%	394 (46.3)	457 (53.7)	0.267
≥75%	114 (50.4)	112 (49.6)	
ASA score			
I–II	269 (50.9)	260 (49.1)	0.234
III–IV	253 (47.2)	283 (52.8)	
Diabetes mellitus			
No	425 (49.5)	433 (50.5)	0.209
Yes	99 (44.8)	122 (55.2)	
Current smoking			
No	425 (48.5)	451 (51.5)	0.948
Yes	99 (48.8)	104 (51.2)	
Obesity			
No	320 (44.0)	407 (56.0)	<0.001
Yes	204 (58.0)	148 (42.0)	
Wound class			
Clean or clean/contaminated	515 (49.0)	535 (51.0)	0.940
Contaminated/dirty	8 (50.0)	8 (50.0)	
Surgical approach			
Minimally invasive	416 (47.3)	463 (52.7)	0.22
Open	140 (43.3)	183 (56.7)	
Improper timing of administration			
No	534 (46.4)	617 (53.6)	0.956
Yes	22 (46.8)	25 (53.2)	
Oral kanamycin			
No	218 (44.0)	278 (56.0)	0.179
Yes	338 (47.9)	368 (52.1)	

CFZ/MTZ = cefazolin/metronidazole group; AMP/SUL = ampicillin/sulbactam group; ASA = American Society of Anesthesiologists; Inappropriate timing of administration = administration cannot be performed within 1 h before surgery; Missing values occur in each variable.

**Table 2 antibiotics-12-01381-t002:** Risk of surgical site infection according to patient factors.

Variables	Surgical Site Infection	Univariate	Multivariate
Antibiotic group			-
CFZ/MTZ (*n* = 556)	21 (55.3)	0.940	
AMP/SUL (*n* = 646)	17 (44.7)	(0.843–1.047)	
Age, in years			-
18–59 (*n* = 405)	11 (28.9)	1.256	
60–99 (*n* = 797)	27 (71.1)	(0.617–2.558)	
Sex			-
Male (*n* = 673)	28 (73.7)	0.444 *	
Female (*n* = 529)	10 (26.3)	(0.214–0.922)	
Site			-
Colon (*n* = 620)	14 (36.8)	1.868	
Rectum (*n* = 580)	24 (63.2)	(0.957–3.648)	
Operation time			-
<75% (*n* = 851)	24 (64.9)	2.103 *	
≥75% (*n* = 226)	13 (35.1)	(1.053–4.199)	
ASA score			-
I–II (*n* = 529)	22 (59.5)	0.663	
III–IV (*n* = 536)	15 (40.5)	(0.340–1.293)	
Diabetes mellitus			-
No (*n* = 858)	28 (75.7)	1.258	
Yes (*n* = 221)	9 (24.3)	(0.585–2.707)	
Current smoking			
No (*n* = 876)	21 (56.8)	3.484 ***	3.489 ***
Yes (*n* = 203)	16 (43.2)	(1.784–6.803)	(1.784–6.822)
Obesity			-
No (*n* = 727)	29 (78.4)	0.560	
Yes (*n* = 352)	8 (21.6)	(0.253–1.237)	
Wound class			-
Clean or clean/contaminated (*n* = 1050)	36 (97.3)	1.878	
Contaminated or dirty (*n* = 16)	1 (2.7)	(0.241–14.606)	
Surgical approach			-
Minimally invasive (*n* = 879)	25 (65.8)	1.433	
Open (*n* = 323)	13 (34.2)	(0.724–2.835)	
Improper timing of administration			
No (*n* = 47)	3 (7.9)	1.433	
Yes (*n* = 1151)	35 (92.1)	(0.724–2.835)	
Oral kanamycin			
No (*n* = 496)	14 (36.8)	1.212	
Yes (*n* = 706)	24 (63.2)	(0.620–2.366)	

CFZ/MTZ = cefazolin/metronidazole group; AMP/SUL = ampicillin/sulbactam group; ASA = American Society of Anesthesiologists; Inappropriate timing of administration = administration cannot be performed within 1 h before surgery; *: *p*-value < 0.05; *****: *p*-value < 0.001.

**Table 3 antibiotics-12-01381-t003:** Comparison of prophylactic effects of CFZ/MTZ and AMP/SUL therapies.

		CFZ/MTZ	AMP/SUL	95% Confidence Interval of the Difference		*p*-Value
		(M ± SD)	(M ± SD)			
Primary endpoint	Surgical site infection	0.038 ± 0.191	0.026 ± 0.160	−0.009	0.032	0.264
Secondary endpoint	Deep/organ space	0.027 ± 0.162	0.012 ± 0.111	−0.001	0.031	0.073
	Percutaneous drainage /surgical intervention	0.016 ± 0.126	0.006 ± 0.079	−0.003	0.020	0.167

CFZ/MTZ = cefazolin/metronidazole group; AMP/SUL = ampicillin/sulbactam group; M = mean; SD = standard deviation.

## Data Availability

The data presented in this study are available on request from the corresponding author. The data are not publicly available due to privacy of patients.

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
