# Peer review of "Comparison of Cefazolin/Metronidazole to Ampicillin/Sulbactam as Preoperative Antibiotics in Colorectal Surgery: A Retrospective, Single-Center Cohort Study"

_antibiotics, 2023, doi:10.3390/antibiotics12091381_

Round 1
Reviewer 1 Report
It is well known that the mechanical preparation of the intestine, aggressive hypothermia and the administration of oxygen during and after surgery are elements that contribute to the incidence of nosocomial infections in the colorectal sphere. No clarifications were made in this regard, in the present study. Other elements that can influence nosocomial infections are prior laparotomy and administration of perioperative transfusion. No clarifications were made in this regard eitherI believe that the biggest limitation of this publication is the exclusion of emergency interventions, this significantly reduces the importance of the results obtained.
Author Response
It is well known that the mechanical preparation of the intestine, aggressive hypothermia and the administration of oxygen during and after surgery are elements that contribute to the incidence of nosocomial infections in the colorectal sphere. No clarifications were made in this regard, in the present study. Other elements that can influence nosocomial infections are prior laparotomy and administration of perioperative transfusion. No clarifications were made in this regard either
I believe that the biggest limitation of this publication is the exclusion of emergency interventions, this significantly reduces the importance of the results obtained.
=>Thank you for your input. None of the patients received intraoperative hypothermic intervention. Additionally, data on laparotomy history, oxygen administration, and blood transfusion could not be obtained. We have added this as a limitation to the study:
”Sixth, data on laparotomy history, oxygen administration, and blood transfusion could not be obtained. These variables should be better controlled in future studies to minimize variability.”
=>Emergency surgery was excluded from this study. Antibiotics for emergency surgery patients are also an important topic. However, it is unclear whether the protocol is followed by patients who usually require emergency surgery. In addition, it is difficult to determine the effectiveness of antibiotics because these patients have various risk factors that are difficult to adjust. So we think Including only elective surgery would help predict more accurate results.
Reviewer 2 Report
The goal of this report was to compare the efficacy between cefazolin/metronidazole and ampicillon/sulbactam as prophylactic antibiotics in colorectal surgery. Authors revealed that there was no significant difference regarding postoperational infection within 30 days in this single-center cohort study. Overall, this study is very interesting and provides valuable insights for communities. The result is well explained and also discussed the potential caveats when interpretation or limitations. I have minor comments:
1. Do you obeserve any side effects associated with the use of the two regimens (CFZ/MTZ vs AMP/SUL) and affect the sugery recovery. Those information will be important as well, in addition to infection rate.
2. Figure 1 is not mentioned in main text, and lacks of figure legend.
Author Response
The goal of this report was to compare the efficacy between cefazolin/metronidazole and ampicillon/sulbactam as prophylactic antibiotics in colorectal surgery. Authors revealed that there was no significant difference regarding postoperational infection within 30 days in this single-center cohort study. Overall, this study is very interesting and provides valuable insights for communities. The result is well explained and also discussed the potential caveats when interpretation or limitations. I have minor comments:
1.Do you obeserve any side effects associated with the use of the two regimens (CFZ/MTZ vs AMP/SUL) and affect the sugery recovery. Those information will be important as well, in addition to infection rate.
=>Side effects of antibiotics were not analyzed in this study because it is generally difficult to distinguish them from postoperative complications. However, there were no serious side effects such as anaphylaxis. In addition, most of the preoperative antibiotics are used at only 1 to 2 doses, so it is expected that side effects will not last long. This was briefly mentioned in the section of limitation.
2.Figure 1 is not mentioned in main text, and lacks of figure legend.
=> Thank you for your comments. We corrected it.
Reviewer 3 Report
Few points which need to be addressed are as mentioned below:
1. Table number 1 needs to be rectified. The total number of patients in CFZ/MTZ group as per table 1 is 609 (318 males+291 females) while the numbers mentioned is 556 in this group. The same is true for many other baseline parameters where the total number depicted in the table does not match with the total numbers mentioned in the respective groups.
2. In results, the authors mention that rectal surgery was more common in CFZ/MTZ group, however, this does not concur with table 1.
3. Please mention few details regarding study setting: such as primary/secondary/tertiary or referral care; any previous data on the incidence of SSIs in the study site; any existence of antibiotic stewardship programme in the study site.
4. Which guidelines are being followed for surgical antibiotic prophylaxis (SAP) in the hospital? OR Is there any hospital antibiotic policy for surgical prophylaxis; If yes, please give details especially with respect to colorectal surgery.
5. The study data was gathered from July 2010 to December 2020. Was any change in empirical surgical antibiotic prophylaxis observed during this decade due to changing resistance patterns?
6. In table 1, the authors mention “improper timing of administration”; please mention clearly, for example, what was the mean time of administration. WHO guidelines recommend administration of SAP within 120 minutes before incision as against the timing as per standard recommendations? Also, what was the duration of SAP?
7. Please include some details on other preoperative and intraoperative measures for prevention of SSIs such as bowel preparation, surgical site preparation, hair removal, infection prevention and control measures etc. as they have a significant impact on incidence of SSIs. Also, was their any change observed in any such protocols during the 10 year study period?
8. Line 111-113; the authors mention low chances of selection bias in the study, which is not clearly described though. What were the factors governing selection of antibiotics from among the two studied groups? What do they mean by saying “antibiotic selection was distributed by period”.
Moderate editing of English language is required.
Author Response
Few points which need to be addressed are as mentioned below:
1.Table number 1 needs to be rectified. The total number of patients in CFZ/MTZ group as per table 1 is 609 (318 males+291 females) while the numbers mentioned is 556 in this group. The same is true for many other baseline parameters where the total number depicted in the table does not match with the total numbers mentioned in the respective groups.
=>Missing value was included in some variables. It made our mistake. We corrected the error.
2.In results, the authors mention that rectal surgery was more common in CFZ/MTZ group, however, this does not concur with table 1.
=> “The rates of rectal surgery and obesity were higher in the CFZ/MTZ group.”
3.Please mention few details regarding study setting: such as primary/secondary/tertiary or referral care; any previous data on the incidence of SSIs in the study site; any existence of antibiotic stewardship programme in the study site.
=> Unfortunately, data on the incidence of SSIs at the institution prior to the study are unavailable. An antibiotic stewardship program has been included in the protocol section of the manuscript.
4.Which guidelines are being followed for surgical antibiotic prophylaxis (SAP) in the hospital? OR Is there any hospital antibiotic policy for surgical prophylaxis; If yes, please give details especially with respect to colorectal surgery.
=> Yes, the antibiotic prophylaxis protocol for colorectal surgery is given below:
“All patients who underwent colorectal surgery during the study period underwent bowel and skin preparation. Antibiotics were administered at least 1 hour prior to skin incision, and re-dosing was conducted 4 hours after the initial dose was administered. No antibiotics were administered after the 24 hours post-surgery. The type of antibiotic used was based on the period, whereby AMP/SUL was used from 2011–2013 and 2016–2020, while CFZ/MTZ was used during the other periods. The infection control unit and infectious medicine specialist monitored the preoperative antibiotic protocols and antibiotic stewardship, while the department of general surgery managed other surgical preparations.”
5.The study data was gathered from July 2010 to December 2020. Was any change in empirical surgical antibiotic prophylaxis observed during this decade due to changing resistance patterns?
=>The causative organisms were only identified in 18 infections, thus making it difficult to analyze changes in resistance over the 10 year period. Additionally, the risk of ESBL-producing Enterobacteriaceae may have increased over time. We discuss this briefly in the discussion section:
“Changes in antibiotic susceptibility should also be considered, as the increasing number of ESBL-producing pathogens is a major problem in nosocomial infections.”
6. In table 1, the authors mention “improper timing of administration”; please mention clearly, for example, what was the mean time of administration. WHO guidelines recommend administration of SAP within 120 minutes before incision as against the timing as per standard recommendations? Also, what was the duration of SAP?
=> Thank you again for your nice comments. The following protocols have been added.
“All patients who underwent colorectal surgery during the study period underwent bowel and skin preparation. Antibiotics were administered at least 1 hour prior to skin incision, and Re-dosing was conducted 4 hours after the initial was administered. No antibiotics were administered after the 24 hours post-surgery mark. The type of antibiotics administered was based on the period, whereby AMP/SUL was used from 2011–2013 and 2016–2020, while CFZ/MTZ was used during the other periods. The infection control unit and infectious medicine specialist monitored the preoperative antibiotic protocols and antibiotic stewardship, while the department of general surgery managed other surgical preparations.”
7. Please include some details on other preoperative and intraoperative measures for prevention of SSIs such as bowel preparation, surgical site preparation, hair removal, infection prevention and control measures etc. as they have a significant impact on incidence of SSIs. Also, was their any change observed in any such protocols during the 10 year study period?
=> Thank you for the above recommendation. We have added the below to the manuscript:
“Fifth, skin and bowel preparations were performed for all patients who underwent colorectal surgery during the study period. Although this preparation is considered not recommended or controversial, it would not influence the results of the study.”
8. Line 111-113; the authors mention low chances of selection bias in the study, which is not clearly described though. What were the factors governing selection of antibiotics from among the two studied groups? What do they mean by saying “antibiotic selection was distributed by period”.
=> In the selection of antibiotics, antibiotic groups were assigned uniformly during a specific period regardless of the characteristics of the physician or patient.
“The type of antibiotics administered was based on the period, whereby AMP/SUL was used from 2011–2013 and 2016–2020, while CFZ/MTZ was used during the other periods.”
Round 2
Reviewer 3 Report
The authors have paid attention to all the queries raised to my satisfaction and have made necessary changes in the manuscript.
Minor editing required.